# Relationship of Cognition and Alzheimer’s Disease with Gastrointestinal Tract Disorders: A Large-Scale Genetic Overlap and Mendelian Randomisation Analysis

**DOI:** 10.3390/ijms232416199

**Published:** 2022-12-19

**Authors:** Emmanuel O. Adewuyi, Eleanor K. O’Brien, Tenielle Porter, Simon M. Laws

**Affiliations:** 1Centre for Precision Health, Edith Cowan University, Joondalup, Perth, WA 6027, Australia; 2Collaborative Genomics and Translation Group, School of Medical and Health Sciences, Edith Cowan University, Joondalup, Perth, WA 6027, Australia; 3Curtin Medical School, Curtin University, Bentley, Perth, WA 6102, Australia

**Keywords:** Alzheimer’s disease, causality, cognition, cognitive traits, educational attainment, gastrointestinal tract disorders, genome-wide association studies (GWAS), local genetic correlation, global genetic correlation, Mendelian randomization

## Abstract

Emerging observational evidence suggests links between cognitive impairment and a range of gastrointestinal tract (GIT) disorders; however, the mechanisms underlying their relationships remain unclear. Leveraging large-scale genome-wide association studies’ summary statistics, we comprehensively assessed genetic overlap and potential causality of cognitive traits and Alzheimer’s disease (AD) with several GIT disorders. We demonstrate a strong and highly significant inverse global genetic correlation between cognitive traits and GIT disorders—peptic ulcer disease (PUD), gastritis-duodenitis, diverticulosis, irritable bowel syndrome, and gastroesophageal reflux disease (GERD), but not inflammatory bowel disease (IBD). Further analysis detects 35 significant (*p* < 4.37 × 10^−5^) bivariate local genetic correlations between cognitive traits, AD, and GIT disorders (including IBD). Mendelian randomisation analysis suggests a risk-decreasing causality of educational attainment, intelligence, and other cognitive traits on PUD and GERD, but not IBD, and a putative association of GERD with cognitive function decline. Gene-based analysis reveals a significant gene-level genetic overlap of cognitive traits with AD and GIT disorders (IBD inclusive, p_binomial-test_ = 1.18 × 10^−3–^2.20 × 10^−16^). Our study supports the protective roles of genetically-influenced educational attainments and other cognitive traits on the risk of GIT disorders and highlights a putative association of GERD with cognitive function decline. Findings from local genetic correlation analysis provide novel insights, indicating that the relationship of IBD with cognitive traits (and AD) will depend largely on their local effects across the genome.

## 1. Introduction

The bidirectional communication between the brain and the gut is well known, and the potential contribution of this phenomenon to the risk of dementia, Alzheimer’s disease (AD), in particular, is a subject of increasing importance that continues to attract major attention in the recent literature [1,2,3,4,5,6,7]. Several studies have reported a positive association between AD (or dementia, generally) and gut microbiota disruption, gastrointestinal tract (GIT) disorders or medications for gastritis, oesophageal reflux disease (GERD), and peptic ulcer disease (PUD) [7,8,9,10,11,12,13,14,15]. The likely roles of the immune (autoimmune) system, dysbiosis, enteric amyloid-beta (*A*β) accumulation, inflammatory processes, vagal nerve stimulation, and lipid metabolism have been suggested [4,5,7,16,17]. However, the mechanisms underlying these relationships require further elucidation.

Clinically, AD is characterised by cognitive deterioration [18,19]. The disorder currently has no known disease-modifying or curative therapies [18,19], largely because the underlying mechanisms are poorly understood. Hence, disentangling the effects of GIT disorders on cognition (and vice versa) could have substantial implications for people with lived experience of AD. For example, such knowledge can enhance a better understanding of the disorders’ poorly understood biological mechanisms, inform preventative or therapy development efforts for AD (and GIT disorders), and provide a basis for further translational studies, especially in light of the gut–brain connection. Importantly, establishing a causal link between genetic predisposition to GIT traits and impaired cognition will not only improve knowledge of their shared underlying biology but can, for instance, suggest treatments for the cure or remission of GIT disorders and/or relevant preventative approaches as potential strategies for slowing cognitive decline.

Similar to the growing evidence linking AD with GIT traits [7,8,9,10,11,12,13,14], emerging findings from conventional observational studies suggest that a range of GIT disorders are associated with cognitive dysfunction [20,21,22,23]. For instance, a recent systematic review and meta-analysis of cross-sectional studies reported a significant risk of cognitive impairment (including deficits in attention, executive function, and working memory) in individuals with inflammatory bowel disease (IBD), compared to controls [20]. Findings in a more recent cross-sectional study also suggest an association of greater subjective and objective cognitive difficulty with higher gastrointestinal symptoms [24]. Similarly, in a multi-site longitudinal study, increasing severity of GIT symptoms was found to be associated with poor performance across all domains of cognition in individuals with Parkinson’s disease [21]. There is also evidence implicating *Helicobacter (H.) pylori*, a known risk for PUD [25] and gut microbiota alterations, in cognitive impairment [1,9,26,27,28]. Interestingly, host genetic susceptibility to *H. pylori* infection in PUD has been demonstrated [29]. Moreover, chronic inflammation, a biological process shared by several GIT disorders, including PUD, GERD, gastritis-duodenitis, and IBD, has been linked with impaired cognition (executive function, especially) [30].

On the other hand, an observational study found no significant association between cognitive function and IBD or irritable bowel syndrome (IBS) [31]. A 2019 systematic review also reported an inconclusive relationship between IBS and several domains of cognitive impairment [32]. Thus, the potential effects of GIT traits on cognition (and vice versa), and their likely underlying mechanisms, remain poorly understood. So far, available evidence on this subject comes from conventional observational studies, which may explain their inconsistent results. In addition to their limitation in drawing causal inferences, observational studies are often susceptible to biases of small sample sizes and confounding influences from many sources, including lifestyle and environment. 

Here, we leveraged large-scale genome-wide association studies (GWAS)’s summary data and utilised a suite of cross-trait statistical genetic analysis methods to explore the genetic relationship between cognitive traits, AD, and GIT disorders. Studies focusing on genetic overlap and causality of one trait on another are critical to an evidence-based understanding of disease mechanisms, therapy characterisation, and potential preventative or treatment development efforts [33]. Hence, unlike the traditional observational studies that are often limited by a range of factors, including reverse causality and residual confounding, the outcomes of the present genetic-based study will provide robust evidence and enhance mechanistic insights into the interplay of GIT disorders with cognition and AD. 

## 2. Results

Figure 1 presents a simplified workflow for this study. First, using the linkage disequilibrium score regression (LDSC) analysis method [34], we assessed and quantified SNP-level pairwise global (genome-wide) genetic correlations between 6 GIT disorders (PUD, GERD, gastritis-duodenitis, IBS, diverticulosis, and IBD), 10 cognitive traits, and AD (Appendix A). Cognitive traits assessed in this study include educational attainment, cognitive performance, intelligence, age of completing full-time education, educational qualifications, fluid intelligence, and four measures of fluid intelligence: chained arithmetic, conditional arithmetic, family relationship calculation, and word interpolation. Second, we assessed local genetic correlations between the GIT disorders, cognitive traits, and AD. Third, we conducted bidirectional two-sample Mendelian randomisation (2SMR) analyses [35] to test for potential causal associations between GIT disorders and cognitive traits. Last, we performed gene-based analysis and subsequently assessed gene-level genetic overlap of three GIT disorders (PUD, GERD, and IBD) and AD with two representative cognitive traits (educational attainment and cognitive performance). 

### 2.1. Global Genetic Correlation of Cognitive Traits with GIT Disorders

Table 1 presents the results of global SNP-based heritability (h^2^_SNP_) estimates (proportion of phenotypic variance that can be attributed to genome-wide common SNPs), obtained using the univariate LDSC analysis method. These estimates ranged from 0.01 to 0.22 for cognitive traits, AD, and GIT disorders. We note, however, that our heritability estimates were based on the observed scale and, thus, may be conservative. 

For example, on the liability scale, the SNP-based global heritability for PUD was 0.06 (se = 0.007), IBS = 0.06 (se = 0.005), and IBD = 0.11 (se = 0.016) [29]. Bivariate LDSC analyses revealed a negative (inverse) and highly significant global genetic correlation (r_g_) between all the cognitive traits and each of the GIT disorders, except IBD (Figure 2 and Appendix A). 

First, LDSC found highly significant negative global genetic correlations between PUD and intelligence (r_g_ = −0.33, se = 0.05, *p* = 2.11 × 10^−11^), cognitive performance (r_g_ = −0.32, se = 0.04, *p* = 9.00 × 10^−16^), and educational attainment (r_g_ = −0.46, se = 0.04, *p* = 5.50 × 10^−33^) [Figure 2a and Appendix A]. Other cognitive traits, including educational qualification, age of completing full-time education, fluid intelligence score, and other measures of fluid intelligence, similarly demonstrated negative and highly significant global genetic correlations with PUD (r_g_ = −0.27–−0.47, se = 0.04–0.10, *p* = 8.41 × 10^−4–^5.75 × 10^−25^) [Figure 2a and Appendix A. Furthermore, we found a positive global genetic correlation between PUD and AD (r_g_ = 0.33, se = 0.12, *p* = 7.55 × 10^−3^), which is consistent with a risk-increasing relationship between the two traits [Figure 2a and Appendix A. These results were based on the unconstrained genetic covariance intercept in the LDSC analysis suggesting, given the evidence of no substantial sample overlap (Appendix A), that the significance of the global correlation estimates was conservative (for example, comparing the results for PUD and AD in Appendix A). 

Second, we observed a similar pattern of strong and even more highly significant negative global genetic correlation between GERD and all the cognitive traits assessed in this study (Figure 2b and Appendix A). For example, GERD demonstrated inverse global genetic correlation with cognitive performance (r_g_ = −0.31, se = 0.02, *p* = 8.35 × 10^−53^), intelligence (r_g_ = −0.31, se = 0.03, *p* = 5.35 × 10^−32^), age of full-time education completion (r_g_ = −0.45, se = 0.03, *p* = 1.25 × 10^−49^), educational attainment (r_g_ = −0.45, se = 0.02, *p* = 1.11 × 10^−151^), and other cognitive traits (r_g_ = −0.24–−0.44, se = 0.02–0.06, *p* = 2.64 × 10^−7^– 8.91 × 10^−92^). Similar to PUD, GERD also demonstrated a positive global genetic correlation with AD (r_g_ = 0.24, se = 0.07, *p* = 2.62 × 10^−4^) [Figure 2b and Appendix A. Notably, these results were also based on the unconstrained genetic covariance intercept. 

Third, LDSC revealed a negative global genetic correlation of other GIT disorders, including gastritis-duodenitis, IBS, and diverticulosis with each of the cognitive traits, and a positive correlation with AD (Figure 2c-e, and Appendix A). It is noteworthy that educational attainment (r_g_ = −0.53, se = 0.03, *p* = 2.84 × 10^−66^) and educational qualifications (r_g_ = −0.54, se = 0.04, *p* = 6.81 × 10^−50^) had the strongest and most significant correlation with gastritis-duodenitis. This pattern of results remained consistent in the correlation of educational attainment with IBS (r_g_ = −0.24, se = 0.03, *p* = 8.54 × 10^−16^) and diverticulosis (r_g_ = −0.24, se = 0.02, *p* = 2.49 × 10^−24^). 

However, IBD behaved differently from the rest of the GIT disorders. We found no evidence of a significant genetic correlation of IBD with AD or any of the cognitive traits except educational attainment (r_g_ = −0.11, se = 0.04, *p* = 3.80 × 10^−3^) and marginally with educational qualification [Figure 2f and Appendix A. Last, using LDSC, we explored the pair-wise global genetic correlation between each of the cognitive traits and AD (without constraining the genetic covariance intercept). As expected, we found a strong and highly significant positive global genetic correlation of cognitive traits with each other and a moderately significant negative genetic correlation with AD. Figure 3 and Appendix A provide details of these results. 

### 2.2. Local Genetic Correlation of Cognitive Traits and AD with GIT Disorders

We performed local genetic correlation analyses of cognitive traits and AD with GIT disorders using LAVA [36]. Unlike the global (genome-wide) correlation analysis using the LDSC method [34], the local approach enabled us to identify and estimate genetic correlation, in specific genomic regions, between GIT disorders, AD, and cognitive traits, thereby providing better insights into their local effects and shared genetic basis. At the threshold of *p* < 4.37 × 10^−5^, adjusting for the total number of tests performed (1144), LAVA detected 35 significant bivariate local genetic correlations, across 14 loci, between GIT disorders (IBD inclusive), cognitive traits, and AD (Table 2). Five of the six GIT disorders, namely, GERD, PUD, gastritis-duodenitis, diverticulosis, and IBD, demonstrate at least one significant local genetic correlation with a cognitive trait or AD at the cut-off value of *p* < 4.37 × 10^−5^. 

GERD had the highest number of local genetic correlations with cognitive traits, accounting for 15 of the 35 (at 7 of the 14 loci) significant bivariate signals detected (Table 2). Local genetic correlations of four GIT disorders (PUD, GERD, gastritis-duodenitis, and diverticulosis) with cognitive traits were in the same direction as their corresponding global genetic correlation. Thus, the findings support the risk-decreasing relationship indicated by the respective LDSC-based global genetic correlations between GIT disorders and cognitive traits. Conversely, IBD showed a positive local genetic correlation with several cognitive traits including educational qualification, cognitive performance, educational attainment, and intelligence, across chr3:47588462–50387742 and chr6:32454578–32539567 (genome build 37, Table 2). This finding is discordant with the (marginally significant) negative global genetic correlation observed between IBD and educational attainment in the LDSC analysis (Figure 2 and Appendix A). At a locus in chromosome 16 (chr16:27443062–29043177), IBD also showed evidence of a negative local genetic correlation with one of the fluid-intelligence measures (FI: chained arithmetic) [Table 2]. Together, the findings may suggest discordant effect directions, across the genome, in the local genetic correlations of IBD with cognitive traits.

Moreover, while the global genetic correlation between GIT disorders and AD were positive, we also found a negative local correlation between GERD and AD at chr19:45040933–45893307 as well as between diverticulosis and AD (at chr6:32454578–32539567). Additionally, we identified two significant positive local genetic correlations between AD and IBD (Table 2). These findings indicate that the shared genetic relationship between some of the GIT disorders (for example IBD, and to a lesser degree, GERD as well as diverticulosis) and AD, across the genome, is not completely positive. Importantly, of the 35 significant bivariate local genetic correlations surviving our threshold (4.37 × 10^−5^), 25 had ‘1′ included in the 95% confidence interval of the variance explained (Table 3), indicating that the local genetic signals of the traits (in those loci) are completely shared, thus further supporting evidence of the shared genetic basis of AD and cognitive traits with GIT disorders.

### 2.3. Results of Causal Relationship Assessment 

We performed bidirectional 2SMR analyses to test for a potential causal association between three GIT disorders and all cognition-related traits. Here, we restricted our analysis to three GIT traits—PUD, GERD, and IBD. We included PUD and GERD as they represent two of the common upper GIT disorders. Importantly, GERD was consistently identified in both the global and local genetic correlation analysis as being strongly associated with cognitive traits and, hence, deserves to be further investigated. Additionally, IBD generally behaved differently from the rest of the other GIT disorders; hence, we consider it important to understand its causal relationship with cognitive traits. 

#### 2.3.1. Causal Relationship of Peptic Ulcer Disease with Cognitive Traits

Our MR analysis indicates a significant causal effect (risk decreasing) of genetic liability to cognitive performance (Odds ratio [OR] = 0.75, 95% confidence interval [95%CI]: 0.67–0.85, *p* = 2.11 × 10^−6^), intelligence (OR = 0.77, 95%CI: 0.69–0.86, *p* = 3.92 × 10^−7^), and educational attainment (OR = 0.56, 95%CI: 0.49–0.63, *p* = 6.68 × 10^−21^) on PUD (Figure 4a and Appendix A). Fluid intelligence score (OR = 0.92, 95%CI: 0.86–0.99, *p* = 1.68 × 10^−2^), educational qualification (OR = 0.74, 95%CI: 0.67–0.83, *p* = 8.49 × 10^−8^), and age of completing full-time education (OR = 0.74, 95%CI: 0.58–0.93, *p* = 1.0 × 10^−2^) were similarly causally associated with a decreased risk of PUD (Figure 4a and Appendix A). These IVW-based results were consistent in at least one additional MR model: weighted median and/or MR-Egger methods (Appendix A), providing more support for the findings. 

Using the MR-PRESSO method, we replicated similar IVW-based significant results, implicating causally protective roles of cognitive performance, intelligence, educational attainment, fluid intelligence score, educational qualification, and age of completing full-time education on the risk of PUD (Table 4 and Appendix A). Outlier corrected results (based on the removal of potentially pleiotropic SNPs) were produced for three of these cognitive traits: intelligence, cognitive performance, and educational attainment (Table 4). The results remained significant even after the corrected analysis. Importantly, findings from the distortion p-value indicate that the presence of the outlier SNPs did not bias the originally estimated causal effects of the cognitive traits on PUD—distortion p-value = 0.84 (intelligence), 0.76 (cognitive performance), and 0.98 (educational attainment). 

Moreover, we further assessed the validity of these causal estimates by conducting additional sensitivity analysis in which we manually checked and excluded potentially pleiotropic SNPs—cognitive traits IVs (exposure) associated with PUD (the outcome variable) at *p*_SNP_ < 0.05. We restricted this sensitivity analysis to the six cognitive traits that showed a significant protective causal influence on PUD: cognitive performance, intelligence, educational attainment, educational qualification, age of completing full-time education, and fluid intelligence score. Notably, our results remained consistent, indicating a risk-decreasing (protective) causal influence of cognitive performance (OR = 0.84, *p* = 1.28 × 10^−3^), intelligence (OR = 0.78, *p* = 5.47 × 10^−6^), educational attainment (OR = 0.68, *p* = 2.73 × 10^−10^), and educational qualification (OR = 0.83, *p* = 2.16 × 10^−4^) on PUD (Appendix A). The weighted median and MR-PRESSO models support these IVW-based findings (Appendix A). Other results, including the MR-Egger intercept and the MR-PRESSO global test, strongly support no evidence of horizontal pleiotropy, increasing confidence in our findings. IVs for this sensitivity analysis and their effects on both the outcome and exposure variables are summarised in Supplementary Tables S7–12.

On the other hand, we found no evidence of a significant causal effect of three other fluid intelligence measures on PUD risk: chained arithmetic (OR = 1.01, 95%CI: 0.90–1.12, *p* = 0.90), family relationship calculation (OR = 0.98, 95%CI: 0.88–1.10, *p* = 0.77), and word interpolation (OR = 0.91, 95%CI: 0.70–1.18, *p* = 0.47) (Table 4, Appendix A and S5). In reverse analyses, in which PUD was assessed as an exposure variable against each of the cognitive traits as an outcome variable, our findings indicate that genetic liability to PUD had no significant causal effect on any of the cognitive traits (Figure 4b and Appendix A). The results were consistent across other MR models, including the weighted median, MR-Egger, and MR-PRESSO (Table 4 and Appendix A). 

#### 2.3.2. Causal Relationship of Gastroesophageal Reflux Disease with Cognitive Traits

Genetic predisposition to six cognitive traits showed evidence of a significant risk-decreasing causal influence on GERD (Figure 5a, Table 5 and Appendix A). These included educational qualification (OR = 0.73, 95%CI: 0.66–0.78, *p* = 3.13 × 10^−17^), intelligence (OR = 0.76, 95%CI: 0.70–0.82, *p* = 1.35 × 10^−12^), and FI score (OR = 0.88, 95%CI: 0.82–0.92, *p* = 1.35 × 10^−6^). Educational attainments (OR = 0.54, 95%CI: 0.50–0.58, *p* = 1.35 × 10^−43^), cognitive performance (OR = 0.69, 95%CI: 0.63–0.73, *p* = 4.20 × 10^−22^), and age of completing full-time education (OR = 0.76, 95%CI: 0.68–0.85, *p* = 9.60 × 10^−7^) similarly showed evidence of a significant causally protective effect on GERD. These results were based on the IVW model and were replicated using other MR models, including the weighted median (all the six named cognitive traits) and MR-Egger (intelligence, FI-score, and cognitive performance) [Appendix A, thereby increasing confidence in these results.

Findings from the MR-PRESSO model also supported the protective causal influence of all six cognitive traits on GERD based on the crude OR estimates (Table 5). By excluding likely pleiotropic variants, the method also produced corrected estimates for five of the cognitive traits, FI-score, cognitive performance, educational attainment, educational qualifications, and intelligence, all of which remained significant even after the corrected analysis. Moreover, the distortion test p-values (Fi-score = 0.69, cognitive performance = 0.65, educational attainment = 0.54, educational qualifications =0.92, and intelligence = 0.99) indicate that the causal estimates before the outlier removal were not biased by the presence of outlier variants. Notably, the FI-score, educational attainment, and cognitive performance assumed even greater significance in their causal relationship with GERD following the MR-PRESSO corrected analysis (Table 5 and Appendix A). 

By changing the direction of analysis, we tested the causal influence of genetic predisposition to GERD on cognitive traits (Figure 5b, Table 5, and Appendix A). The findings reveal a significant causal association of GERD with cognitive function decline in educational attainment (OR = 0.86, 95%CI: 0.81–0.93, *p* = 4.56 × 10^−4^), cognitive performance (OR = 0.90, 95%CI: 0.83—0.97, *p* = 1.84 × 10^−2^), and educational qualifications (OR = 0.86, 95%CI: 0.77—0.94, *p* = 5.07 × 10^−3^). Similar results of a significant decrease in cognitive function were found in the FI-score (OR = 0.75, 95%CI: 0.63—0.91, *p* = 2.45 × 10^−3^), age of completing full-time education (OR = 0.83, 95%CI: 0.71—0.93, *p* = 3.93 × 10^−3^), intelligence (OR = 0.89, 95%CI: 0.83—0.94, *p* = 4.52 × 10^−4^), and fluid intelligence measure of word interpolation (OR = 0.73, 95%CI: 0.61—0.85, *p* = 2.31 × 10^−4^). These IVW-based findings were largely consistent across the weighted median and MR-PRESSO models (Table 5 and Appendix A). MR-PRESSO produced corrected causal estimates for the relationship of five cognitive traits (educational attainment, fluid intelligence score, intelligence, educational qualifications, and age of completing full-time education) with GERD, all of which retained their significance.

To further test the robustness of these findings, we undertook additional sensitivity analysis and manually excluded GERD SNPs (exposure variable) associated with the corresponding cognitive traits (outcome variables) at *p*_SNP_ < 0.05. This manual exclusion of potentially pleiotropic SNPs complements the MR-PRESSO analysis (which also excludes potentially pleiotropic SNPs), and we found that the significance of our estimates waned and was no longer evident following the analysis. We note, however, that GERD’s IVs were substantially low, which may contribute to the nonsignificant results. We followed up on this analysis by relaxing the IV selection cut-off point to the genome-wide suggestive level (*p* < 1 × 10^−5^) for GERD. Using these new IVs, we first repeated all the MR analysis procedures, and findings were consistent with our previous results, indicating a significant inverse causal association of GERD with educational attainment, intelligence, educational qualification, age of completing full-time education, fluid intelligence score, and cognitive performance. These IVW-based results were consistent across the weighted median and the MR-PRESSO models. Following a manual removal of IVs with *p*_SNP_ < 0.05, we found nominally significant estimates of GERD’s association with educational attainment, intelligence, and educational qualification, both in the IVW and the MR-PRESSO models (Appendix A). Findings were significant across the IVW, the weighted median, and MR-PRESSO for the causal effect of GERD on the age of completing full-time education. Together, these findings are more supportive of a putative causality of GERD with decreased cognitive function than otherwise. GERD’s IVs (exposure variable, at genome-wide suggestive level) utilised in this follow-up analysis and their effects on the outcome variables (cognitive traits) are summarised in Supplementary Tables S15–19).

#### 2.3.3. Causal Relationship of Inflammatory Bowel Disease with Cognitive Traits

We tested the causal association between IBD and cognitive traits, first with cognitive traits as the exposure variables. We found no evidence of a significant causal effect of cognitive traits on IBD (Figure 6a). The results for this analysis were consistent across the IVW, weighted median, MR-Egger, and MR-PRSSO analyses (Table 6 and Appendix A). We reversed the direction of the analysis and tested IBD as the exposure and cognitive traits as outcome variables and similarly found no evidence of a causal effect of IBD on cognitive traits. Figure 6a,b, Table 6, and Appendix A provide details of these findings.

### 2.4. Results of Gene-Level Genetic Overlap Analysis

We performed gene-based analysis to further assess the genetic overlap, at the gene level, of cognitive traits with GIT disorders. Given the strong SNP-based genetic correlation between each of the cognitive traits included in this study (Figure 3 and Appendix A), we restricted our gene-based analysis to only two of the cognitive traits—educational attainment and cognitive performance. We tested the relationship of these two cognitive traits with PUD, GERD, IBD, and, for comparison, AD. 

In the first set of analyses, we used a total of 7,091,604 SNPs that were overlapping between the educational attainment and PUD GWASs in performing equivalent gene-based analysis for the respective traits. This analysis resulted in a total of 18,650 protein-coding genes for each of educational attainment and PUD GWAS. At *p*_gene_ < 0.05, a total of 1511 genes were associated with PUD and 6762 with educational attainment, while a total of 626 genes overlapped between the two traits (Table 7). To assess gene-level genetic overlap, we compared the expected proportion of gene overlap, at *p*_gene_ < 0.05, with the observed proportion of gene overlap (see methods for details). The results of the exact binomial test support a significant gene-level genetic overlap between educational attainment and PUD at the *p*_gene_ < 0.05 threshold (*p*_binomial-test_ = 3.85 × 10^−4^) (Table 7). For example, the observed proportion of gene overlap between the educational attainment and the PUD GWAS (9.2%) was significantly greater than the null (8.1%) [*p*_binomial-test_ = 3.85 × 10^−4^], indicating a significant gene-level overlap between the two traits (Table 7). 

In the second set of analyses, we utilised a total of 7091,610 SNPs overlapping cognitive performance and PUD GWAS in conducting gene analysis, producing 18,650 protein-coding genes for the traits. Of these, 5273 and 1511 genes were associated with cognitive performance and PUD, respectively, at *p*_gene_ < 0.05. Findings from the exact binomial test similarly supported a significant gene-level genetic overlap between the two traits at *p*_gene_ < 0.05 (*p*_binomial-test_ = 1.18 × 10^−3^), indicating that the observed proportion of gene overlap was more than expected by chance. 

Following a similar process of analysis, we found a significant gene-level genetic overlap between GERD and each of educational attainment (*p*_binomial-test_ = 2.20 × 10^−16^) and cognitive performance (*p*_binomial-test_ = 2.20 × 10^−16^) [Table 7]. We also found a significant gene-level genetic overlap of IBD with educational attainment (*p*_binomial-test_ = 3.95 × 10^−6^) and cognitive performance (*p*_binomial-test_ = 2.13 × 10^−5^). These results are notably consistent with the relationship between AD and cognitive traits (Table 7). For example, the observed proportion of genes overlapping the educational attainment and the AD GWAS (11.2%) was significantly higher than the expected proportion (9.6%) [*p*_binomial-test_ = 7.79 × 10^−6^, Table 7]. 

## 3. Discussion

Support for the involvement of the gut–brain link in the risk of AD continues to gather momentum [1,2,3,4,5,6,7]. However, emerging evidence on the connection between cognition and GIT traits is inconsistent [20,21,22,23,31,32]. To advance our understanding on the relationship of cognitive traits and AD with GIT disorders, the present study analysed several large-scale GWAS’ summary data using well-regarded statistical genetic methods. Our findings reveal a significant negative global genetic correlation between all cognitive traits and GIT disorders, including PUD, GERD, gastritis-duodenitis, IBS, and diverticulosis, and, together with the results of gene-level overlap, support evidence of their shared genetic signatures. 

Furthermore, MR analysis reveals a protective causal effect of genetically predicted cognitive traits on the risk of GIT disorders (PUD and GERD). Thus, consistent with the protective effects previously reported for other disorders, including AD and coronary heart disease [37,38], these findings indicate that cognitive traits such as intelligence and higher educational attainments can contribute to a reduced risk of GIT disorders. In a reverse MR analysis, GERD demonstrated a significant putative causal association with decreased cognitive function. This finding suggests that genetic predisposition to GERD contributes to an increased risk of cognitive function decline, which may partly explain the previous findings of a positive association and genetic correlation between GERD and AD or dementia [7,11]. Although in more conservative sensitivity testing, the significance of this finding waned and was no longer evident, together, the results of our follow-up analysis are more supportive of a putative causal effect of GERD on cognitive traits than otherwise. We note, however, that this finding (genetic overlap of GERD with a decline in cognitive performance) may only partially contribute to causality, given that environmental factors could also play an important role in this regard. For example, individuals with a high level of cognition are likely to be more aware of healthy lifestyles, which may lower their risk of GIT disorders. Conversely, lower quality of life in individuals with GIT disorders may also contribute to impaired cognitive functions. 

Given that local genetic effects can deviate substantially from the average represented by findings in the LDSC-based global genetic correlation estimates [36], we performed local genetic correlation analyses between cognitive traits, AD, and GIT disorders. Findings from this analysis reveal significant local genetic correlations (at specific genomic locations) between several cognitive traits and GIT disorders. Notably, the effect direction of the estimated correlations was concordant across four of the GIT disorders (PUD, GERD, gastritis-duodenitis, and diverticulosis), supporting the negative (risk-decreasing) relationship indicated in their respective global genetic correlation with cognitive traits. Interestingly, IBD behaved differently from other GIT disorders. For example, we found no significant global genetic correlation between IBD and cognitive traits (except marginally with educational attainment and educational qualification). Similarly, and consistent with a recent genetic analysis [7], IBD showed no evidence of a significant genetic correlation with AD. A logical explanation for this finding would be the comparatively smaller cases (and lower effective sample size) of the IBD GWAS [7]. 

However, our local genetic correlation estimates provide new insights into this relationship, revealing discordant effects in the local genomic associations of IBD with cognitive traits, suggesting a likely counteraction of opposing genetic effects and, hence, a lack of (or weakened) significant overall effects in the global genetic estimates. Supportive of this position, our gene-level genetic overlap analysis robustly identified significant genetic overlap between GIT disorders (IBD inclusive) and cognitive traits, indicating evidence of shared genetics between IBD and cognitive traits. While the gene-based method cannot discriminate the effect direction of the overlap estimates, it can indicate whether two or more traits share a genetic basis and, thus, is informative, as in the present study. Together, our findings support a significant genetic overlap between IBD and cognitive traits (as confirmed by our gene-level genetic overlap analysis) but with some levels of discordant effects at different loci (suggested in our local correlation findings), which may explain the minimal signal detected in the LDSC-based global genetic correlation estimates. Thus, whereas some observational studies suggest a positive association of IBD with AD or cognitive impairment [12,20,39], our genetic-based study indicates that the nature of this relationship depends on effects at specific genomic loci. This observation may explain the lack of significant causality of IBD with cognitive traits and AD [7] and/or the inconsistency reported in some observational studies [20,31,39,40,41]. Therefore, making a biological sense of the relationship between IBD and cognitive traits (and by extension AD) will require a focus on the effects at the specific genomic locus.

Our findings have considerable implications for practices and further studies. First, educational attainment, a potentially modifiable factor, had the strongest and most significant genetic correlation as well as a protective causal association with several of the GIT disorders. Indeed, IBD, which generally was not correlated with cognitive traits, showed at least a nominally significant negative global genetic correlation with educational attainment. These findings support education as a possible avenue for reducing the risk of GIT disorders: for example, by encouraging higher educational attainment or a possible increase in the length of schooling [37,38]. Previous evidence demonstrating strong (bidirectional) relationships, and indeed, a causal association of intelligence with educational attainment [37] supports this premise. Hence, policy efforts aimed at increasing educational attainment (or cognitive training) may contribute to a higher level of intelligence with expected consequences for better health outcomes, including a reduced risk of GIT disorders. 

Second, GERD demonstrated evidence of a putative causal association with a decline in cognitive function across many cognitive traits assessed in this study. While, to the best of our knowledge, this is the first study to suggest this causal relationship, this finding generally agrees with other studies, especially a recent longitudinal study that reported an increased incidence of dementia with GERD [11]. Thus, the GIT disorder may be a risk factor for cognitive impairment, supporting the importance of probing or investigating signs or symptoms of cognitive dysfunction in patients presenting with GERD. This suggestion may potentially benefit early detection of cognitive decline and, hence, the provision of appropriate intervention(s) towards reducing the rate of cognitive decline. Third, given current findings, more studies are needed to investigate whether treatment for cure or remission of GERD can contribute to a reduced risk of cognitive decline. The importance of this recommendation comes to the fore in light of conflicting observational evidence concerning the association of dementia with medications for treating GERD [11,15]. Hence, future studies are needed to clarify whether a cure or remission of GERD has any relationship with cognition or dementia. 

Last, one of the notable findings in our local genetic correlation analysis pertains to the somewhat discordant local effects of IBD in its relationship with cognitive traits. This finding may explain the lack of significant global genetic correlation observed between these traits in the present study, as well as between IBD and AD in a recent GWAS-based study [7]. Thus, while IBD shares a genetic relationship with cognitive traits (as confirmed by our gene-level genetic overlap analysis), the global relationship was not evident, which may be because of discordance in their effect direction across the genome. This finding brings new insight into the relationship of IBD with cognitive traits (and AD), which may shape the direction of future studies. For example, some risk genes for AD may be protective against IBD, and vice versa.

The major strength of our study is the use of well-regarded statistical genetic approaches in assessing the relationship of cognitive traits and AD with GIT disorders. These approaches are based on the use of genetic variants inherited before disease manifestation or exposure to environmental or lifestyle confounders. Thus, our genetic-based study is less susceptible to limitations of the traditional observational studies, including reverse causation and residual confounding—providing robust evidence on this subject. Importantly, we used multiple and complementary methods which provide a comprehensive evaluation of the relationship between cognitive traits and GIT disorders. To the best of our knowledge, this is the first study to assess the relationship of cognitive traits and AD systematically and comprehensively with GIT disorders using the statistical genetic approach. 

Nevertheless, our study has some limitations that should be considered while interpreting its findings. First, the GWAS data utilised were of individuals of European ancestry; hence, readers need to exercise caution in comparing or generalising findings to those of other ancestries. Second, while our analyses generally indicate no substantial sample overlap between GIT disorders and cognitive traits, granted GWAS data were majorly from the UKB, unknown sample overlap can slightly inflate our results but is not expected to alter our conclusions. Given this observation, we did not constrain the genetic covariance intercept in our global genetic correlation analysis. Similarly, we provided estimates of potential sample overlap for use in local genetic correlation analysis, meaning findings from these analyses could not have been influenced by sample overlap bias. Third, MR analysis is based on stringent assumptions that may be difficult to meet, especially in highly polygenic traits such as those assessed in this study. To minimise chances of pleiotropic or other potential biases, we followed several precautionary measures, excluded pleiotropic variants (where applicable) and used additional MR models in our study. These measures, especially the use of additional MR models, enhance confidence in our findings. Last, although we found no evidence for a significant causal effect of PUD on cognitive traits (unlike with GERD), we cannot rule out this possibility. The GWAS for PUD has comparatively limited IVs, which may contribute to present results; hence, future studies are needed to further clarify this relationship as more powerful PUD GWAS becomes available.

## 4. Materials and Methods

### 4.1. Data Sources

Well-powered GWAS summary statistics sourced from international research consortia or publicly available databases/repositories were analysed in this study. We utilised GWAS summary data for 10 cognition-related traits, including intelligence (sample size [*n*] = 269,867) [42], cognitive performance (*n* = 257,828) [43], educational qualification (*n* = 318,526) [44], educational attainment (*n* = 766,345) [43], age of completing full-time education (*n* = 253,580) [44], and fluid intelligence (FI) score (*n* = 125,935) [44]. Other measures of fluid intelligence including FI word interpolation (*n* = 124,929) [44], FI chained arithmetic (*n* = 68,065) [44], FI conditional arithmetic (*n* = 96,994) [44], and FI family relationship calculation (*n* = 99,934) [44] were also included to provide insights into their relationship with GIT disorders. 

Furthermore, we analysed GWAS summary data of 6 GIT disorders, including PUD (cases = 16,666, controls = 439,661), IBD (cases = 7045, controls = 449,282), and irritable bowel syndrome (IBS, cases = 28,518, controls = 426,803), all of which were sourced from a recently published GIT GWAS [29]. The GWAS for GERD (cases = 71,522, controls = 261,079), comprising data from the QSKIN study and the United Kingdom biobank (UKB) [45], was also analysed in our study. Additional GIT GWAS, from the Lee Lab (https://www.leelabsg.org/resources; accessed on 14 June–26 September 2022), including gastritis-duodenitis (GD, cases = 28,941, controls = 378,124) and diverticular disease (cases = 27,311, controls = 334,783) were similarly included in the present study. 

Lastly, we utilised one of the largest publicly available AD GWAS, comprising 71,880 cases and 383,378 controls, made up of both clinically diagnosed AD and AD-by-proxy [46]. A summary of all these GWAS data is provided in Table 1 and Figure 1. Additional specific details about the data, including, where applicable, links for downloading them, are provided in Appendix A. A more comprehensive description of the GWAS and their quality control procedures are available in the relevant publications referenced in this study. All the GWAS summary data analysed were from individuals of European descent. 

### 4.2. Genome-Wide (Global) Genetic Correlation Analysis

We performed LDSC regression analyses, first, to estimate h^2^_SNP_ for each of the traits included in our study and, second, to assess the global (genome-wide) genetic correlation of GIT disorders with AD and cognitive traits. LDSC [34] is a well regarded method for estimating SNP-based heritability and genetic correlation between two or more traits using GWAS summary data (https://github.com/bulik/ldsc; accessed on 14 June–26 September 2022). In the present study, we used the standalone version of the software and followed the scripts provided by the program developers (https://github.com/bulik/ldsc/wiki/Heritability-and-Genetic-Correlation; accessed on 14 June–26 September 2022). SNP-based heritability (univariate LDSC) was estimated on the observed scale, and in all analyses, we used the precomputed European population LD scores from the 1000G for HapMap 3 SNPs. 

We performed a cross-trait genetic correlation between GIT disorders (PUD, GERD, gastritis-duodenitis, IBS, diverticulosis, and IBD) and each of the 10 cognitive traits and AD, using the bivariate LDSC analysis approach. LDSC can adjust for sample overlap when the genetic covariance intercepts are not constrained [34]. Although the estimated genetic covariance intercept did not suggest a substantial sample overlap between any of the pair of traits analysed here (Appendix A), we did not constrain the intercepts in our analysis; thus, the significance of our correlation estimates can be considered conservative (Appendix A). We considered a bivariate correlation result significant at *p* < 0.005 (0.05/10), a Bonferroni adjustment for testing 10 cognitive traits, and nominally significant at *p* < 0.05.

### 4.3. Local Genetic Correlation Analysis

Global genetic estimates, obtained, for example, using the LDSC method, capture the average genetic correlation of two traits across the entire genome. However, this approach can mask the local genetic effects and may not detect a significant correlation if effect directions are discordant. To address this limitation and gain insight into regions contributing disproportionately to the genetic correlation of cognitive traits and GIT disorders, we also performed local genetic correlation analyses. In the present study, we utilised a newly developed method for assessing local genetic correlation—the LAVA (local analysis of covariant association) [36]. This analytic tool has additional advantages, including its ability to simultaneously model more than two traits and perform conditional or partial genetic correlation assessment between a phenotype and predictor traits [36]. 

In brief, we implemented LAVA using the R statistical platform (in the Unix environment) and estimated bivariate local genetic correlation between GIT disorders (PUD, GERD, gastritis-duodenitis, IBS, diverticulosis, and IBD), AD, and 10 cognitive traits across the genome. First, as part of the input files required in LAVA [36], we estimated potential sample overlap between the traits being assessed using the LDSC approach [34]. In this instance, we assessed the global genetic correlation between the traits without constraining the genetic covariance intercept [34,36]. The original locus definition file provided by the program developers was used in our study. Additionally, we utilised the Phase 3 EUR 1000G as reference data, having a matching ancestry (European) with the GWAS datasets included in the present study. 

To prevent false positive results, the direction of effect needs to be consistent across all phenotypes. Hence, LAVA first extracts SNPs common to the GWAS summary data and aligns their effect alleles to the reference data before commencing analysis [36]. SNPs that could not be aligned were excluded from analyses. Second, we conducted a LAVA-based univariate analysis to estimate local genetic heritability for each of the GIT disorders, AD, and cognitive traits. The presence of a sufficient signal in the local univariate analysis is required for the subsequent detection of meaningful bivariate genetic correlation in LAVA [36]. Hence, using the results of the univariate analysis, traits were selected for bivariate analysis at *p* < 5 × 10^−5^. Third, we performed pairwise bivariate local genetic correlation analysis across the genome for all the traits included in our study but prioritised the relationship of cognitive traits and AD with GIT disorders, the focus of the present study. Applying a Bonferroni correction for the number of bivariate tests performed, we selected significant bivariate loci at *p* < 4.37 × 10^−5^ (that is 0.05/1144).

### 4.4. Bidirectional Mendelian Randomisation Analysis

We performed bidirectional 2SMR analyses to assess the potential causal effects of genetic predisposition to GIT disorders on cognitive traits and vice versa. Mendelian randomisation (MR) is an instrumental variable-based analytic method that incorporates genetics into epidemiological study designs, thereby, mimicking randomised control trials (RCTs) and providing a reliable approach for estimating causality between one trait and another—considered as ‘exposure’ and ‘outcome’ variables [47]. The method utilises existing genetic data and, thus, is time and cost-effective. 

In the present study, we conducted MR analyses using the standalone 2SMR package (version 0.5.6, https://mrcieu.github.io/TwoSampleMR/; accessed on 14 June–26 September 2022), implemented in R (version 4.0.2). We first assessed GIT disorders as the exposure against each of the cognitive traits as the outcome. In a reverse analysis, we also assessed each of the cognitive traits as an exposure against GIT disorders as outcome variables. Linkage disequilibrium (LD) independent (at r^2^ < 0.001) genome-wide significant (*p* < 5 × 10^−8^) SNPs, robustly associated with the exposure data, were utilised as instrumental variables (IVs). This stringent threshold corresponds to an F-statistics of about 30 [33], which ensures that the IVs are strong (F-statistic > 10), thereby minimising the likelihood of weak instrument bias. Two cognitive traits (FI: chained arithmetic, FI: conditional arithmetic) had too few (or no) genome-wide significant SNPs; therefore, we relaxed the IVs selection cut-off point to the genome-wide suggestive level (*p* < 1 × 10^−5^) for these traits. 

Next, we extracted the exposure IVs from the outcome data and carried out data harmonisation to ensure that SNP effects corresponded to the same allele for both exposure and outcome data. Variants that could not be harmonised or those that had intermediate allele frequencies were excluded. We implemented the inverse variance weighted (IVW) approach, it being the most powerful model for detecting causal association in MR analysis when all instruments are valid [48]. In this instance, the effects of genetic variants on the outcome variables were regressed on those of the exposure variables, weighted by the inverse of their variance. The IVW model assumes no horizontal pleiotropy among the IVs. We assessed a potential violation of this assumption using the MR-Egger intercept. It is expected that the MR-Egger intercept will not deviate significantly from zero where the assumption of no unbalanced pleiotropy holds. 

To ensure that our findings were not driven primarily by individual influential variants, we assessed the possibility of this occurrence using the ‘leave-one-out’ approach and performed individual MR analyses. Furthermore, we applied two additional MR models, the weighted median and the MR-Egger regression methods, for sensitivity testing and to complement the IVW model in causality assessment. The weighted median and the MR-Egger models are more robust to genetic heterogeneity and can provide valid estimates even when up to 50% (the weighted median) or all (the MR-Egger) of the IVs are invalid [49,50]. We assessed SNP effects’ heterogeneity using Cochran’s Q statistics approach and prioritised the results of the weighted median model where there was evidence for significant heterogeneity. 

Finally, we estimated causality by excluding likely pleiotropic variants (where present), using another MR model, the MR-PRESSO (Mendelian randomisation pleiotropy residual sum and outlier) analysis [51]. The MR-PRESSO detects and excludes pleiotropic variants and recalculates causal estimates (corrected MR estimates) [51]. Where MR-PRESSO produced corrected results, we assessed whether the presence of outlier IVs biased the crude causal estimates using the distortion test p-values, and where this was the case, we prioritised MR-PRESSO’s corrected estimates.

### 4.5. Gene-Level Genetic Overlap Assessment

We extended our study beyond the SNP level to gene-based analysis, thereby furthering the assessment of the genetic overlap of cognitive traits with GIT disorders. Genes are more closely related to biology than SNPs and, thus, can provide more interpretable results while enhancing insights into the relationship between cognitive traits and GIT disorders. To estimate gene-level genetic overlap, we performed gene-based analyses and assessed whether the observed proportion of genes overlapping GIT disorders and the corresponding cognitive traits was more than expected by chance, similar to practices in previous studies [52,53,54,55,56,57]. 

Firstly, we conducted gene-based analysis separately for cognitive traits and GIT disorders using the multi-marker analysis of genomic annotation (MAGMA v1.08) software [58] (in the FUMA [59] online platform v1.3.8, accessed June 14–Sept 26, 2022). To ensure that equivalent gene-based tests were performed, we utilised only SNPs that overlapped between the respective cognitive traits and GIT disorders GWAS (educational attainment and PUD, for example) in the gene-based analysis. We used the EUR 1000G Phase 3 data as a reference panel, and SNPs were assigned to genes in MAGMA at the window size of ‘±0 kb’. 

Secondly, using the respective outputs of the gene-based analysis for GIT disorders and cognitive traits, we extracted genes associated with each of the traits at *p*_gene_ < 0.05 for subsequent use in assessing gene-level genetic overlap. At this threshold, we identified the total number of genes for GIT disorders and the respective cognitive traits GWAS. Additionally, we identified the total number of genes overlapping between the pair of traits. Finally, we compared the expected proportions of gene overlap (the null) with the observed proportion of gene overlap using the exact binomial statistical test. 

The expected proportion of gene overlap was defined as the number of genes associated with GIT disorders (for example, PUD) at *p*_gene_ < 0.05 divided by the total number of genes obtained in the gene-based analysis for the GIT disorder (PUD, for instance) [52,53,54,55,56,57]. The observed proportion of genes was defined as the ratio of the number of overlapping genes (between PUD and educational attainment GWAS, for example) to the total number of genes associated with the cognitive trait (educational attainment, for instance) at *p*_gene_ < 0.05 [52,53,54,55,56,57]. To test the statistical difference between the expected and the observed proportion of gene overlap, we applied a one-sided exact binomial test implemented in the R statistical platform, thereby determining whether the observed proportion of gene overlap was more than the null. For a significant gene-level genetic overlap, the observed proportion of overlap must be statistically more than the null. 

## 5. Conclusions

Our study identified highly significant risk-decreasing global and local genetic correlations of cognitive traits with GIT disorders, including PUD, GERD, gastritis-duodenitis, IBS, and diverticulosis (but largely not with IBD in the global genetic correlation analysis), providing new insights into the relationships of these traits. Findings from MR analysis highlight causally protective roles of cognitive traits on GIT disorders (PUD and GERD but not IBD). Together, these findings support the protective roles of cognitive traits such as intelligence and higher educational attainments in the risk of GIT disorders (PUD and GERD), indicating they could contribute to a reduced risk of the disorders. On the other hand, GERD was putatively associated with decreased cognitive functions and may, thus, be a risk factor for cognitive impairment. Further analysis, based on local genetic correlation, reveals discordant local effects of IBD with one or more cognitive traits, providing new insights into its relationships and suggesting that the relationship of IBD with cognitive traits (and AD) will depend largely on their local effects across the genome. Overall, current findings support educational attainments as a potential avenue for reducing the risk of GIT disorders. Hence, encouraging higher educational attainment and/or cognitive training may be beneficial in reducing the risk of GIT disorders.

## Figures and Tables

**Figure 1 ijms-23-16199-f001:**
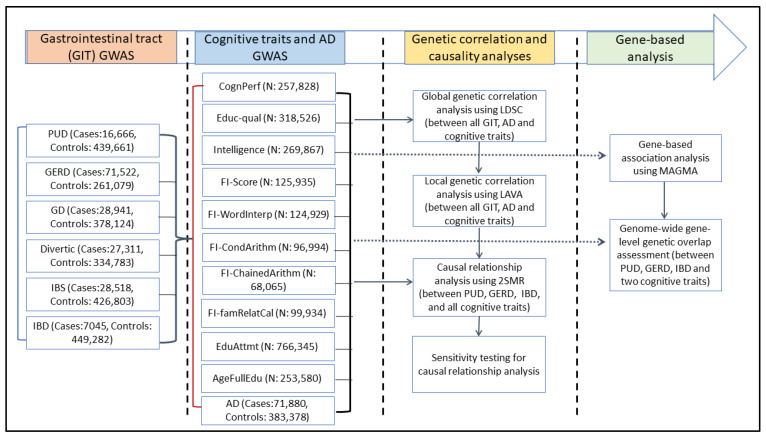
Study design and workflow: examining the relationship of cognitive traits and AD with GIT disorders. AD: Alzheimer’s disease, IBS: irritable bowel syndrome, PUD: peptic ulcer disease, GERD: gastroesophageal reflux disease, IBD: inflammatory bowel disease, GD: gastritis-duodenitis, Divertic: diverticulosis, AgeFullEdu: age completed full-time education, FI-ChainedArithm: fluid intelligence-chained arithmetic, FI-CondArithm: fluid intelligence-conditional arithmetic, FI-famRelatCal: fluid intelligence-family relationship calculation, FI-WordInterp: fluid intelligence-word interpolation, CognPerf: cognitive performance, Educ-qual: educational qualification, EduAttmt: educational attainment. GWAS: genome-wide association studies, GIT: gastrointestinal tract, LDSC: linkage disequilibrium score regression, LAVA: local analysis of [co]variant association, MAGMA: multi-marker analysis of genomic annotation, 2SMR: two-sample Mendelian randomisation.

**Figure 2 ijms-23-16199-f002:**
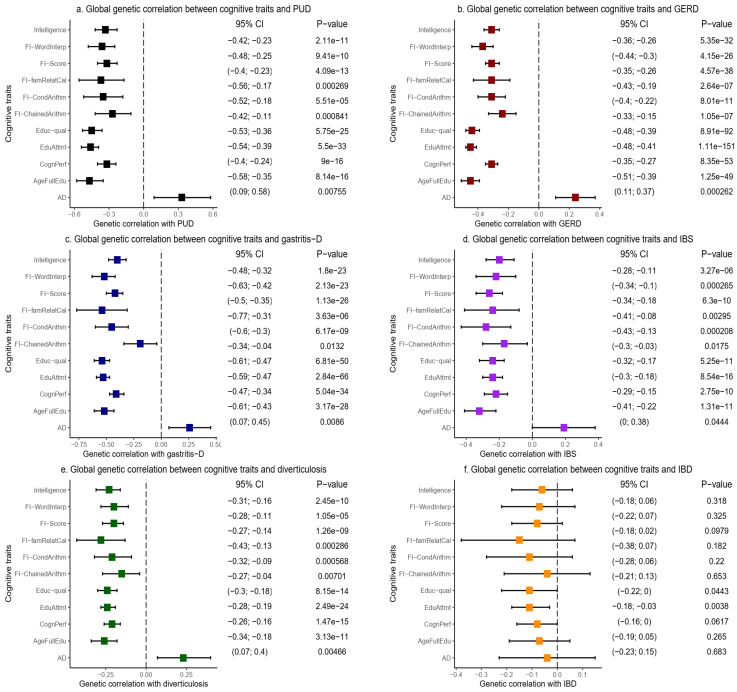
Global genetic correlation of cognitive traits and AD with GIT disorders using LDSC. AD: Alzheimer’s disease, IBS: irritable bowel syndrome, PUD: peptic ulcer disease, GERD: gastroesophageal reflux disease, IBD: inflammatory bowel disease, Gastritis-D: gastritis-duodenitis, AgeFullEdu: age completed full-time education, FI-ChainedArithm: fluid intelligence-chained arithmetic, FI-CondArithm: fluid intelligence-conditional arithmetic, FI-famRelatCal: fluid intelligence-family relationship calculation, FI-WordInterp: fluid intelligence-word interpolation, CognPerf: cognitive performance, Educ-qual: educational qualification, EduAttmt: educational attainment. GIT: gastrointestinal tract, LDSC: linkage disequilibrium score regression.

**Figure 3 ijms-23-16199-f003:**
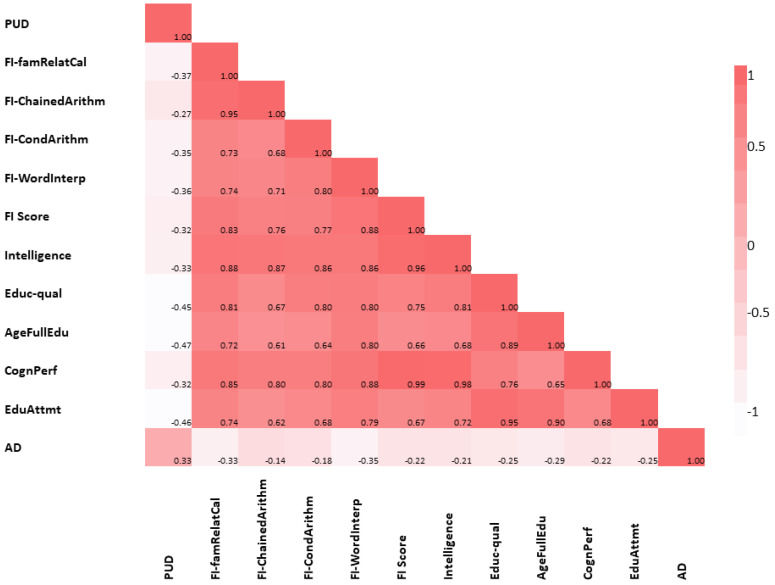
Heatmap of pair-wise global genetic correlation of cognitive traits, AD, and PUD using the LDSC approach. AD: Alzheimer’s disease, PUD: peptic ulcer disease, AgeFullEdu: age completed full-time education, FI-ChainedArithm: fluid intelligence-chained arithmetic, FI-CondArithm: fluid intelligence-conditional arithmetic, FI-famRelatCal: fluid intelligence-family relationship calculation, FI-WordInterp: fluid intelligence-word interpolation, CognPerf: cognitive performance, Educ-qual: educational qualification, EduAttmt: educational attainment. LDSC: linkage disequilibrium score regression.

**Figure 4 ijms-23-16199-f004:**
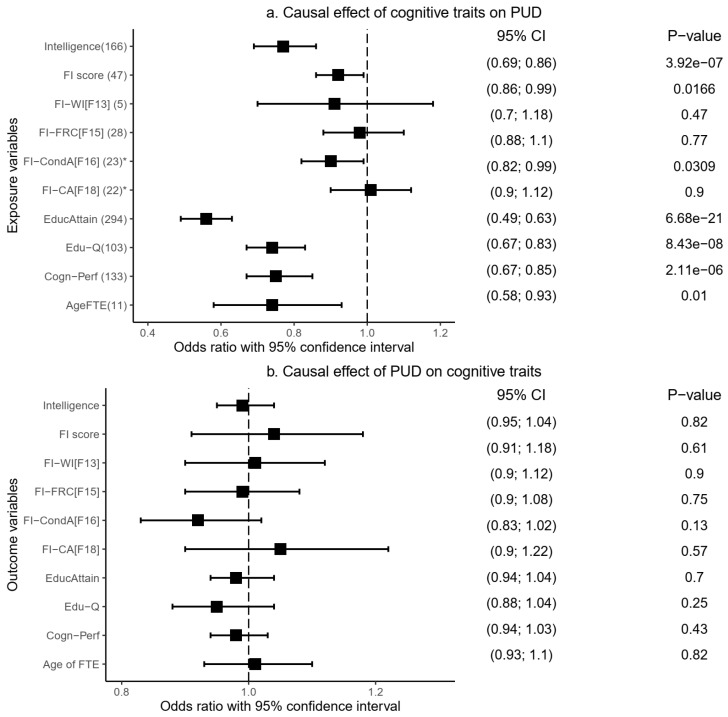
Causal relationship between cognitive traits and PUD based on the IVW MR model. PUD: peptic ulcer disease, Age of FTE: age completed full-time education, FI-CA[F18]: fluid intelligence-chained arithmetic, FI-CondA[F16]: fluid intelligence-conditional arithmetic, FI-FRC[F15]: fluid intelligence-family relationship calculation, FI-WI[F13]: fluid intelligence-word interpolation, Cogn-Perf: cognitive performance, EducAttain: educational attainment, * instrumental variables selected at *p*_snp_ < 1 × 10^−5^, ivw: inverse variance weighted, MR: Mendelian randomisation.

**Figure 5 ijms-23-16199-f005:**
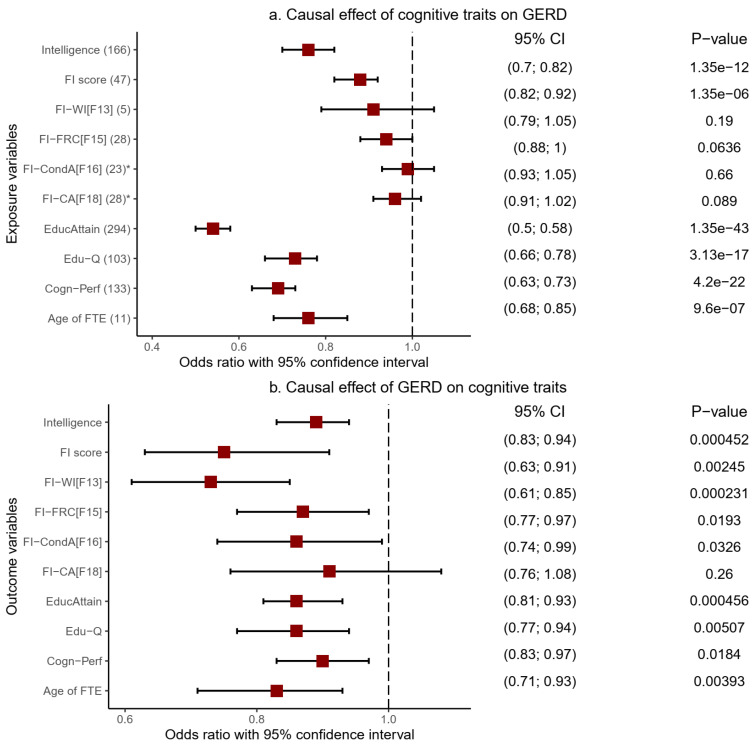
Causal relationship between cognitive traits and GERD based on the IVW MR model. GERD: gastroesophageal reflux disease, Age of FTE: age completed full-time education, FI-CA[F18]: fluid intelligence-chained arithmetic, FI-CondA[F16]: fluid intelligence-conditional arithmetic, FI-FRC[F15]: fluid intelligence-family relationship calculation, FI-WI[F13]: fluid intelligence-word interpolation, Cogn-Perf: cognitive performance, EducAttain: educational attainment, * instrumental variables selected at *p*_snp_ < 1 × 10^−5^, ivw: inverse variance weighted, MR: Mendelian randomisation.

**Figure 6 ijms-23-16199-f006:**
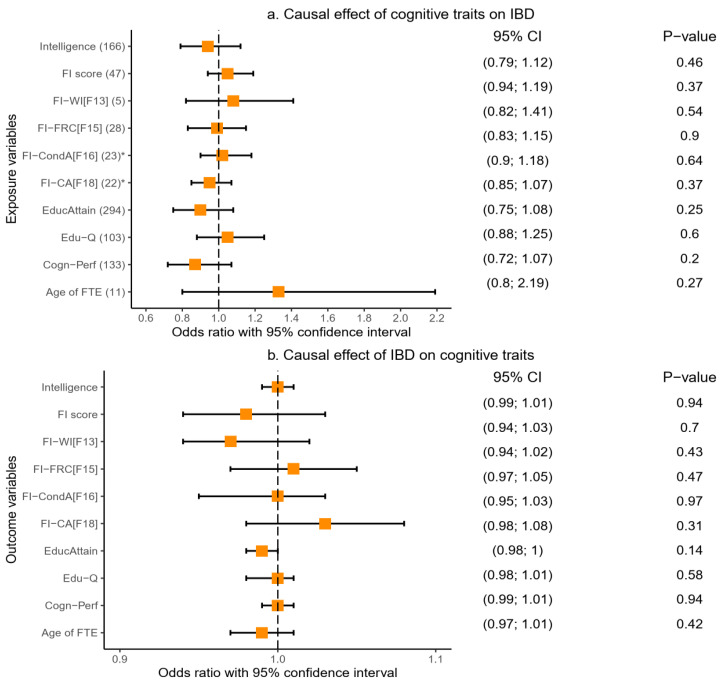
Causal relationship between cognitive traits and IBD based on the IVW MR model. IBD: inflammatory bowel disease, Age of FTE: age completed full-time education, FI-CA[F18]: fluid intelligence-chained arithmetic, FI-CondA[F16]: fluid intelligence-conditional arithmetic, FI-FRC[F15]: fluid intelligence-family relationship calculation, FI-WI[F13]: fluid intelligence-word interpolation, Cogn-Perf: cognitive performance, EducAttain: educational attainment, * instrumental variables selected at *p*_snp_ < 1 × 10^−5^, ivw: inverse variance weighted, MR: Mendelian randomisation.

**Table 1 ijms-23-16199-t001:** Summary of the GWAS data analysed with heritability estimates.

Phenotype Class	Phenotype Name	Sample Size	Global h^2^_SNP_ (observed scale)	Global h^2^_SNP_ SE	Atlas or Source ID (Study and Year)	Ancestry
**Cognitive traits**	Cognitive performance	257,828	0.20	0.01	4067 (Lee et al., 2018)	European
	Intelligence	269,867	0.20	0.01	3785 (Savage et al., 2018)	
	Fluid intelligence (FI) score	125,935	0.22	0.01	3413 (Watanabe et al., 2019)	
	FI test–FI3: word interpolation	124,929	0.07	0.01	3402 (Watanabe et al., 2019)	
	FI test–FI6: conditional arithmetic	96,994	0.05	0.01	3404 (Watanabe et al., 2019)	
	FI test–FI8: chained arithmetic	68,065	0.07	0.01	3406 (Watanabe et al., 2019)	
	FI test–FI5: family relationship calculation	99,934	0.03	0.01	3403 (Watanabe et al., 2019)	
	Educational attainment	766,345	0.11	0.00	4066 (Lee et al., 2018)	
	Education–Qualifications	318,526	0.11	0.00	3409 (Watanabe et al., 2019)	
	Age completed full-time education	253,580	0.05	0.00	3203 (Watanabe et al., 2019)	
**Alzheimer’s disease (AD)**	AD	455,258	0.01	0.00	(Jansen et al., 2019)	
**Gastrointestinal tract (GIT) disorders**	Peptic ulcer disease (PUD)	456,327	0.01	0.00	(Wu et al., 2021)	
	Gastroesophageal reflux disease (GERD)	332,601	0.07	0.00	(An et al., 2019)	
	Gastritis-duodenitis (GD)	407,065	0.02	0.00	Phecode 535 (Lee lab)	
	Diverticulosis	362,094	0.04	0.00	Phecode 562 (Lee lab)	
	Irritable bowel syndrome (IBS)	455,321	0.01	0.00	(Wu et al., 2021)	
	Inflammatory bowel disease (IBD)	456,327	0.01	0.00	(Wu et al., 2021)	

FI: fluid intelligence, SNP: single nucleotide polymorphism, SE: standard error, h2: heritability.

**Table 2 ijms-23-16199-t002:** LAVA local genetic correlations between GIT disorders, AD, and cognitive traits.

Locus	Chr	Start	Stop	Phenotype 1	Phenotype 2	Parameter	R2	*p*
1	3	47588462	50387742	GERD	Educational attainment	−0.87	0.76	3.14 × 10^−9^
	3	47588462	50387742	GERD	FI-Score	−0.98	0.96	6.63 × 10^−7^
	3	47588462	50387742	GERD	Intelligence	−0.83	0.69	8.72 × 10^−7^
	3	47588462	50387742	GERD	Educational qualification	−0.76	0.58	3.68 × 10^−6^
	3	47588462	50387742	GERD	Cognitive performance	−0.82	0.67	5.26 × 10^−6^
	3	47588462	50387742	GERD	Age of fulltime education	−0.95	0.91	2.69 × 10^−5^
	3	47588462	50387742	IBD	Educational qualification	0.85	0.72	5.81 × 10^−8^
	3	47588462	50387742	IBD	Cognitive performance	0.89	0.80	2.83 × 10^−7^
	3	47588462	50387742	IBD	Educational attainment	0.70	0.49	4.57 × 10^−7^
	3	47588462	50387742	IBD	Intelligence	0.75	0.56	3.45 × 10^−6^
2	6	27261036	28666364	GERD	Cognitive performance	−0.78	0.61	4.50 × 10^−6^
3	6	31106494	31250556	Diverticulosis	Age of fulltime education	−1.00	1.00	2.29 × 10^−5^
4	6	32454578	32539567	IBD	AD	1.00	1.00	4.80 × 10^−7^
	6	32454578	32539567	IBD	Educational attainment	0.77	0.59	1.14 × 10^−5^
	6	32454578	32539567	Diverticulosis	Educational qualification	−1.00	1.00	4.81 × 10^−10^
	6	32454578	32539567	Diverticulosis	Educational attainment	−1.00	1.00	5.02 × 10^−9^
	6	32454578	32539567	Diverticulosis	Age of fulltime education	−1.00	1.00	1.51 × 10^−7^
	6	32454578	32539567	Diverticulosis	AD	−0.92	0.85	8.93 × 10^−6^
5	6	32539568	32586784	IBD	AD	0.99	0.98	1.10 × 10^−8^
6	6	98173004	99678876	GERD	Educational attainment	−0.60	0.36	1.65 × 10^−6^
	6	98173004	99678876	PUD	Cognitive performance	−0.69	0.47	5.90 × 10^−7^
	6	98173004	99678876	PUD	Intelligence	−0.57	0.32	1.51 × 10^−5^
	6	98173004	99678876	PUD	FI-Score	−0.64	0.41	1.91 × 10^−5^
7	11	112755447	113889019	GERD	Educational attainment	−0.87	0.75	1.34 × 10^−6^
8	13	58245844	59751795	GERD	Educational attainment	−0.65	0.42	7.79 × 10^−6^
	13	58245844	59751795	GERD	Cognitive performance	−0.83	0.69	1.12 × 10^−5^
9	14	22760701	23985936	GD	Cognitive performance	−0.81	0.66	1.05 × 10^−5^
	14	22760701	23985936	GD	Age of fulltime education	−1.00	1.00	4.22 × 10^−5^
10	15	96864279	98025684	GD	Educational qualification	−0.60	0.36	3.49 × 10^−5^
11	16	27443062	29043177	IBD	FI Chained arithmetic	−0.77	0.59	1.30 × 10^−7^
12	16	53393883	54866095	GD	Cognitive performance	−0.72	0.51	1.23 × 10^−5^
	16	53393883	54866095	GERD	Intelligence	−0.52	0.27	3.22 × 10^−6^
	16	53393883	54866095	GERD	Cognitive performance	−0.57	0.33	7.08 × 10^−6^
13	17	45883902	47516224	GERD	Cognitive performance	−0.80	0.65	1.16 × 10^−5^
14	19	45040933	45893307	GERD	AD	−0.40	0.16	3.78 × 10^−5^

Chr: chromosome, PUD: peptic ulcer disease, GERD: gastroesophageal reflux disease, IBD: inflammatory bowel disease, GD: gastritis-duodenitis, AD: Alzheimer’s disease, FI-Chained arithmetic: fluid intelligence-chained arithmetic, *p*: *p*-value, LAVA: local analysis of [co]variant association.

**Table 3 ijms-23-16199-t003:** Overview of significant bivariate local correlations and proportions.

Phenotype 1	Phenotype 2	N. Sig.	CI97.5 = 1	Percentage
GERD	Cognitive performance	5	4	80%
GERD	Educational attainment	4	2	50%
GERD	Intelligence	2	1	50%
IBD	Educational attainment	2	1	50%
Diverticulosis	Age of fulltime education	2	2	100%
IBD	AD	2	2	100%
GD	Cognitive performance	2	2	100%
GERD	FI-Score	1	1	100%
GERD	Educational qualification	1	1	100%
GERD	Age of fulltime education	1	1	100%
IBD	Educational qualification	1	1	100%
IBD	Cognitive performance	1	1	100%
IBD	Intelligence	1	1	100%
Diverticulosis	Educational qualification	1	1	100%
Diverticulosis	Educational attainment	1	1	100%
Diverticulosis	AD	1	1	100%
PUD	Cognitive performance	1	0	0%
PUD	Intelligence	1	0	0%
PUD	FI-Score	1	0	0%
GD	Age of fulltime education	1	1	100%
GD	Educational qualification	1	0	0%
IBD	FI-Chained arithmetic	1	1	100%
GERD	AD	1	0	0%

PUD: peptic ulcer disease, GERD: gastroesophageal reflux disease, IBD: inflammatory bowel disease, GD: gastritis-duodenitis, AD: Alzheimer’s disease, FI-Chained arithmetic: fluid intelligence-chained arithmetic, *p*: *p*-value. The table summarises the number of significant bivariate local genetic correlations with the proportion of significant loci for which the 95%CI included 1 (CI^97.5^ = 1).

**Table 4 ijms-23-16199-t004:** Results of bidirectional MR-PRESSO analysis for PUD and cognitive traits.

Exposure (nSNPs)	Outcome	MR-PRESSO RESULTS					MR-Egger Intercept	
		Global test P	Raw OR	*p*	Cor-OR	*p*	Intercept	*p*
**Cognitive traits (exposure) and PUD (outcome)**								
Age of fulltime education (11)	PUD	0.11	0.74	**2.6 × 10^−2^**	-	-	0.035	0.13
Educational qualification (103)	PUD	0.001	0.73	**5.26 × 10^−7^**	-	-	−0.01	0.06
Intelligence (166)	PUD	0.002	0.77	**7.84 × 10^−6^**	0.76	**1.90 × 10^−6^**	−0.0042	0.43
FI Chained arithmetic (22) *	PUD	0.0048	1.01	0.91	-	-	−0.0035	0.8
FI Cond arithmetic (23) *	PUD	0.26	0.9	**4.2 × 10^−2^**	-	-	−0.015	0.5
FI−famRelatCal (28)	PUD	0.09	0.98	0.78	-	-	−0.0028	0.76
Fluid intelligence score (47)	PUD	0.144	0.92	**2.0 × 10^−2^**	-	-	−0.0025	0.81
FI−Word interpolation (5)	PUD	0.071	0.91	0.51	-	-	0.1008	0.6
Cognitive performance (133)	PUD	0.0052	0.74	**5.40 × 10^−6^**	0.77	**9.55 × 10^−6^**	0.00077	0.89
Educational attainment (294)	PUD	<0.001	0.56	**1.90 × 10^−18^**	0.55	**3.85 × 10^−19^**	−0.0017	0.6
**PUD (exposure) and cognitive traits (outcome)**								
PUD (7)	Age of fulltime education	0.063	1.01	0.83	-	-	−0.0043	0.86
PUD (7)	Educational qualification	2.0 × 10^−4^	0.94	0.3	0.97	0.18	0.013	0.61
PUD (7)	Intelligence	0.0098	0.99	0.83	0.99	0.8	0.0098	0.49
PUD (7)	FI Chained arithmetic	0.22	1.05	0.59	-	-	−0.0077	0.86
PUD (7)	FI Cond arithmetic	0.82	0.92	0.066	-	-	−0.014	0.64
PUD (7)	FI−famRelatCal	0.41	0.99	0.76	-	-	0.0027	0.91
PUD (7)	Fluid intelligence score	0.0164	1.04	0.63	1.04	0.48	0.035	0.37
PUD (7)	FI−Word interpolation	0.548	1.01	0.88	-	-	0.0069	0.83
PUD (7)	Cognitive performance	0.0176	0.98	0.47	0.998	0.93	0.014	0.32
PUD (7)	Educational attainment	<2.0 × 10^−4^	0.98	0.73	0.97	0.32	0.015	0.33

nSNPs: number of SNPs utilised as instrumental variables, SNP: single nucleotide polymorphism, PUD: peptic ulcer disease, FI Chained arithmetic: fluid intelligence-chained arithmetic, FI Cond arithmetic: fluid intelligence-conditional arithmetic, FI−famRelatCal: fluid intelligence-family relationship calculation, FI−Word interpolation: fluid intelligence-word interpolation, IVW: inverse variance weighted, *p*: *p*-value, MR-PRESSO: Mendelian randomisation pleiotropy residual sum and outlier, OR: odds ratio, *p*: *p*-value, Cor-OR: corrected odds ratio, *instrumental variables were selected at *p_snp_* < 1 × 10^−5^. Note spaces marked with a dash indicate that there were no outlier SNPs, and hence, there were no outlier corrected results in the MR-PRESSO analysis.

**Table 5 ijms-23-16199-t005:** Results of bidirectional MR-PRESSO analysis between GERD and cognitive traits.

Exposure (nSNPs)	Outcome	MR-PRESSO RESULTS					MR-Egger Intercept	
		Global Test P	Raw OR	*p*	Cor-OR	*p*	Intercept	*p*
**Cognitive traits (exposure) and GERD (outcome)**								
Age of fulltime education (11)	GERD	0.26	0.76	**6.24 × 10^−4^**	-	-	−0.0045	0.70
Educational qualification (103)	GERD	<2 × 10^−4^	0.72	**2.94 × 10^−14^**	0.72	**2.23 × 10^−13^**	−0.0061	0.18
Intelligence (166)	GERD	<2 × 10^−4^	0.75	**8.67 × 10^−12^**	0.76	**3.75 × 10^−11^**	0.0065	0.08
FI Chained arithmetic (22) *	GERD	0.23	0.96	0.1	-	-	−0.0052	0.35
FI Cond arithmetic (23) *	GERD	0.041	0.99	0.65	-	-	−0.012	0.017
FI−famRelatCal (28)	GERD	0.018	0.94	7.46 × 10^−2^	-	-	−0.0031	0.57
Fluid intelligence score (47)	GERD	<2 × 10^−4^	0.88	**1.53 × 10^−5^**	0.87	**3.43 × 10^−7^**	0.016	0.04
FI−Word interpolation (5)	GERD	0.036	0.91	0.26	0.85	**2.31 × 10^−2^**	0.021	0.84
Cognitive performance (133)	GERD	<2 × 10^−4^	0.69	**4.86 × 10^−17^**	0.67	**9.48 × 10^−20^**	0.0056	0.12
Educational attainment (294)	GERD	<0.001	0.54	**6.94 × 10^−34^**	0.53	**8.56 × 10^−45^**	−0.0036	0.12
**GERD (exposure) and cognitive traits (outcome)**								
GERD (19)	Age of fulltime education	2 × 10^−4^	0.83	**9.89 × 10^−3^**	0.80	**2.77 × 10^−3^**	−0.02	0.26
GERD (19)	Educational qualification	<2 × 10^−4^	0.86	**3.84 × 10^−2^**	0.90	**1.18 × 10^−2^**	−0.012	0.37
GERD (19)	Intelligence	<2 × 10^−4^	0.89	**2.51 × 10^−3^**	0.91	**2.29 × 10^−3^**	−0.01	0.22
GERD (19)	FI Chained arithmetic	0.56	0.91	0.25	-	-	−0.026	0.22
GERD (19)	FI Cond arithmetic	0.49	0.86	0.046	-	-	−0.027	0.13
GERD (19)	FI−famRelatCal	0.62	0.87	**2.26 × 10^−2^**	-	-	−0.0095	0.52
GERD (19)	Fluid intelligence score	<2 × 10^−4^	0.75	**7.23 × 10^−3^**	0.79	**1.15 × 10^−2^**	−0.032	0.15
GERD (19)	FI−Word interpolation	0.19	0.73	**1.70 × 10^−3^**	-	-	−0.041	0.04
GERD (19)	Cognitive performance	<2 × 10^−4^	0.90	**2.99 × 10^−2^**	0.95	0.11	−0.016	0.13
GERD (19)	Educational attainment	<2 × 10^−4^	0.87	**2.53 × 10^−3^**	0.88	**3.05 × 10^−4^**	−0.0077	0.41

nSNP: number of SNPs utilised as instrumental variables, SNP: single nucleotide polymorphism, GERD: gastroesophageal reflux disease, FI Chained arithmetic: fluid intelligence-chained arithmetic, FI Cond arithmetic: fluid intelligence-conditional arithmetic, FI−famRelatCal: fluid intelligence-family relationship calculation, FI−Word interpolation: fluid intelligence-word interpolation, IVW: inverse variance weighted, *p*: *p*-value, MR-PRESSO: Mendelian randomisation pleiotropy residual sum and outlier, OR: odds ratio, Cor-OR: corrected odds ratio, * instrumental variables were selected at *p_snp_* < 1 × 10^−5^. Note spaces marked with a dash indicate that there was no outlier corrected results in the MR-PRESSO analysis.

**Table 6 ijms-23-16199-t006:** Results of bidirectional MR-PRESSO analysis for IBD and cognitive traits.

Exposure (nSNPs)	Outcome	MR-PRESSO RESULTS					MR-Egger Intercept	
		Global Test P	Raw OR	*p*	Cor-OR	*p*	Intercept	*p*
**Cognitive traits (exposure) and IBD (outcome)**								
Age of fulltime education (11)	IBD	2 × 10^−4^	1.33	0.30	1.06	0.73	−0.019	0.72
Educational qualification (103)	IBD	<2 × 10^−4^	1.05	0.61	0.98	0.81	−0.024	0.023
Intelligence (166)	IBD	4 × 10^−4^	0.94	0.46	0.96	0.61	−0.0094	0.26
FI Chained arithmetic (22) *	IBD	0.60	0.95	0.34	-	-	0.00088	0.94
FI Cond arithmetic (23) *	IBD	0.66	1.03	0.60	-	-	0.0065	0.60
FI−famRelatCal (28)	IBD	0.23	0.99	0.89	-	-	0.0051	0.70
Fluid intelligence score (47)	IBD	0.006	1.06	0.36	1.02	0.70	0.0032	0.87
FI−Word interpolation (5)	IBD	0.36	1.08	0.59	-	-	0.30	0.13
Cognitive performance (133)	IBD	4 × 10^−4^	0.88	0.20	0.87	0.11	0.0053	0.55
Educational attainment (294)	IBD	0.0018	0.90	0.25	0.85	0.067	−0.0051	0.32
**IBD (exposure) and cognitive traits (outcome)**								
IBD (25)	Age of fulltime education	0.091	0.99	0.43	-	-	−0.00060	0.90
IBD (25)	Educational qualification	0.0046	1.00	0.60	0.99	0.23	0.0014	0.68
IBD (25)	Intelligence	0.006	1.00	0.94	1.00	0.61	0.00086	0.73
IBD (25)	FI Chained arithmetic	0.077	1.03	0.31	-	-	−0.0066	0.51
IBD (25)	FI Cond arithmetic	0.178	1.00	0.97	-	-	0.0073	0.34
IBD (25)	FI−famRelatCal	0.027	1.01	0.48	-	-	0.0071	0.35
IBD (25)	Fluid intelligence score	0.0012	0.99	0.72	-	-	0.0055	0.45
IBD (25)	FI−Word interpolation	0.24	0.98	0.43	-	-	0.0013	0.85
IBD (25)	Cognitive performance	2 × 10^−4^	1.00	0.94	1.00	0.70	5.0 × 10^−5^	0.99
IBD (25)	Educational attainment	0.0228	0.99	0.15	0.99	1.29 × 10^−2^	0.00068	0.63

nSNP: number of SNPs utilised as instrumental variables, SNP: single nucleotide polymorphism, IBD: inflammatory bowel disease, FI Chained arithmetic: fluid intelligence-chained arithmetic, FI Cond arithmetic: fluid intelligence-conditional arithmetic, FI−famRelatCal: fluid intelligence-family relationship calculation, FI−Word interpolation: fluid intelligence-word interpolation, IVW: inverse variance weighted, *p*: *p*-value, MR-PRESSO: Mendelian randomisation pleiotropy residual sum and outlier, OR: odds ratio, Cor-OR: corrected odds ratio, * instrumental variables were selected at *p*_snp_ < 1 × 10^−5^. Note spaces marked with a dash indicate that there were no outlier SNPs, and hence, there were no outlier corrected results in the MR-PRESSO analysis.

**Table 7 ijms-23-16199-t007:** Gene-level genetic overlap of cognitive traits with PUD, GERD, IBD, and AD.

Discovery Set			Target Set			No. of Genes Overlapping the Discovery and the Target Sets at *p*_gene_ < 0.05	Proportion of Gene Overlap		Binomial Test
PUD, GERD, IBD or AD	Total no. of Discovery Set (PUD, GERD, IBD or AD) Genes	No. of Discovery Set Genes at *p*_gene_ < 0.05	Cognitive Traits	Total No. of Target set (Cognitive Traits) Genes	No. of Target Set Genes at *p*_gene_ < 0.05		Expected	Observed	*p* Value
PUD	18,650	1511	Educational attainment	18,650	6761	625	0.081	0.092	3.85 × 10^−4 *^
			Cognitive performance	18,650	5273	489	0.081	0.093	1.18 × 10^−3^
GERD	18,729	3290	Educational attainment	18,729	6832	1752	0.176	0.255	2.20 × 10^−16^
			Cognitive performance	18,729	5285	1336	0.176	0.253	2.20 × 10^−16^
IBD	18,650	1920	Educational attainment	18,650	6761	811	0.103	0.20	3.95 × 10^−6^
			Cognitive performance	18,650	5273	636	0.103	0.121	2.13 × 10^−5^
AD	18,865	1813	Educational attainment	18,865	6720	753	0.096	0.112	7.79 × 10^−6^
			Cognitive performance	18,865	5212	591	0.096	0.113	1.94 × 10^−5^

AD: Alzheimer’s disease, GERD: gastroesophageal reflux disease, PUD: peptic ulcer disease, IBD: inflammatory bowel disease, No.: number. ^*^ Result explained: We compared the expected proportions of gene overlap (the null) with the observed proportion of gene overlap. The expected proportion of gene overlap = number of genes associated with the discovery set (PUD) at *p*_gene_ < 0.05 (1511) / total number of genes (18,650). Observed proportion of genes = number of overlapping genes (625)/total number of genes associated with the discovery set (Educational attainment) at *p*_gene_ < 0.05 (6761). To test whether the observed proportion of genes is more than expected by chance, we applied a one-sided exact binomial test implemented in the R statistical platform [binom.test(625,6761,0.0810, alternative = c(“greater”))].

## Data Availability

The GWAS data analysed in this study were sourced from public repositories and research groups or consortia, as described in the section for data sources. More specific details about these data and, where applicable, links to their sources are summarised in Appendix A. Data generated in the present study are included in this published article [and its Appendix A].

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
