# Peer review of "Relationship of Cognition and Alzheimer’s Disease with Gastrointestinal Tract Disorders: A Large-Scale Genetic Overlap and Mendelian Randomisation Analysis"

_ijms, 2022, doi:10.3390/ijms232416199_

Round 1

Reviewer 1 Report

Here, thes workers have leveraged large-scale genome-wide association studies (GWAS) summary statistics and assessed genetic overlap and potential causality of impaired cognitive performance and Alzheimer’s disease (AD) with several gastrointestinal tract (GIT) disorders. They have found a significant inverse genetic correlation between cognitive traits and peptic ulcer disease (PUD), gastritis-duodenitis, diverticulosis, irritable bowel syndrome, and gastroesophageal reflux (GERD) but not inflammatory bowel disease (IBD). There were 35 significant bivariate local genetic correlations between cognitive traits, AD, and GIT disorders including IBD. Mendelian randomisation analysis suggested educational attainment, intelligence, and other cognitive traits decreased risk of PUD, and GERD but not IBD, and GERD was associated with cognitive function decline. It is concluded that the findings support the protective roles of higher educational attainments and cognitive performance on the risk of GIT disorders.

The study is of interest but the observed genetic overlap between impaired cognitive performance and GIT disorders is only partially likely to contribute to causality. It seems highly likely that environmental factors also play a role here  - presumably subjects with a high level of cognition are likely to be more aware of healthy life styles to lower risk of some these GIT disorders. Conversely if you have GIT disorders they may well impair your cognitive performance due to a lower quality of life.

Author Response

Reviewer 1

Comment: Here, these workers have leveraged large-scale genome-wide association studies (GWAS) summary statistics and assessed genetic overlap and potential causality of impaired cognitive performance and Alzheimer’s disease (AD) with several gastrointestinal tract (GIT) disorders. They have found a significant inverse genetic correlation between cognitive traits and peptic ulcer disease (PUD), gastritis-duodenitis, diverticulosis, irritable bowel syndrome, and gastroesophageal reflux (GERD) but not inflammatory bowel disease (IBD). There were 35 significant bivariate local genetic correlations between cognitive traits, AD, and GIT disorders including IBD. Mendelian randomisation analysis suggested educational attainment, intelligence, and other cognitive traits decreased the risk of PUD, and GERD but not IBD, and GERD was associated with cognitive function decline. It is concluded that the findings support the protective roles of higher educational attainments and cognitive performance on the risk of GIT disorders.

The study is of interest but the observed genetic overlap between impaired cognitive performance and GIT disorders is only partially likely to contribute to causality. It seems highly likely that environmental factors also play a role here - presumably, subjects with a high level of cognition are likely to be more aware of healthy lifestyles to lower the risk of some of these GIT disorders. Conversely, if you have GIT disorders, they may well impair your cognitive performance due to a lower quality of life.

Response: We thank the reviewer for finding merits in our work. We agree with the reviewer’s comment concerning our results on the putative association and causality of GERD with a decline in cognitive functions. We have made corrections to indicate that our finding of genetic overlap of GIT disorders with a decline in cognitive performance may only partially contribute to causality, as in the case of GERD. Notably, environmental factors may also play an important role in this regard.  For example, individuals with a high level of cognition are likely to be more aware of healthy lifestyles which may lower their risks of these GIT disorders. Conversely, lower levels of quality of life, in individuals with GIT disorders, may contribute to impaired cognitive functions.

Lines 482 – 482 (of the revised manuscript): “We note, however, that this finding (genetic overlap of GERD with a decline in cognitive performance) may only partially contribute to causality given environmental factors may also play an important role in this regard. For example, individuals with a high level of cognition are likely to be more aware of healthy lifestyles which may lower their risks of the GIT disorder. Conversely, lower levels of quality of life, in individuals with GIT disorders, may also contribute to impaired cognitive functions.”

Reviewer 2 Report

In this study Adewuyi and colleagues follow up their published analysis of possible correlation between Alzheimer’s disease (AD) and Gastrointestinal Tract (GIT) disorders (Comm. Biology 2022; https://doi.org/10.1038/s42003-022-03607-2). Similar to their previous work, here the Authors perform deep statistical (global and genetic) analysis to assess the possible relationship between cognitive traits, (AD) and GIT disorders and found: i) a significant negative global genetic correlation between cognitive traits and GIT disorders; ii) a protective causal effect of the genetically predicted cognitive traits on the risk of GIT disorders; and iii) for GIT disorders, a significant putative causal association with decreased cognitive function.

The statistical analysis is sound and the results sufficiently well described. Also, no evident overlap with their previous publication is evident albeit, across the two papers, topics and methodological approaches are very similar and cohorts of original data are at least partly shared (see table 1 of this Manuscript vs Table 1 in the published paper).

Very minor: use of commas might be improved here and there..

Altogether, in my opinion the present Ms is acceptable for publication in IJMS in the present form.

Author Response

Reviewer 2

Comment: In this study, Adewuyi and colleagues follow up their published analysis of the possible correlation between Alzheimer’s disease (AD) and Gastrointestinal Tract (GIT) disorders (Comm. Biology 2022; https://doi.org/10.1038/s42003-022-03607-2). Similar to their previous work, here the Authors perform deep statistical (global and genetic) analysis to assess the possible relationship between cognitive traits, (AD) and GIT disorders and found: i) a significant negative global genetic correlation between cognitive traits and GIT disorders; ii) a protective causal effect of the genetically predicted cognitive traits on the risk of GIT disorders; and iii) for GIT disorders, a significant putative causal association with decreased cognitive function.

The statistical analysis is sound and the results are sufficiently well described. Also, no evident overlap with their previous publication is evident albeit, across the two papers, topics and methodological approaches are very similar and cohorts of original data are at least partly shared (see table 1 of this Manuscript vs Table 1 in the published paper).

Very minor: use of commas might be improved here and the…

Altogether, in my opinion, the present Ms is acceptable for publication in IJMS in the present form.

Response: We thank the reviewer for finding our work to be acceptable for publication. We carried out a more critical read to ensure minor grammatical issues, including the use of punctuation marks, are corrected. The manuscript was revised for grammatical accuracy.